# Ethical Considerations for Echinoderms: New Initiatives in Welfare

**DOI:** 10.3390/ani13213377

**Published:** 2023-10-31

**Authors:** Augusto César Crespi-Abril, Tamara Rubilar

**Affiliations:** 1Instituto Patagónico del Mar (IPaM), Universidad Nacional del a Patagonia San Juan Bosco (UNPSJB), Boulevard Brown 2915, Puerto Madryn 9120, Argentina; 2Laboratorio de Oceanografía Biológica (LOBio), Centro Para el Estudio de Sistemas Marinos (CESIMAR–CONICET), Boulevard Brown 2915, Puerto Madryn 9120, Argentina; rubilar@cenpat-conicet.gob.ar; 3Laboratorio de Química de Organismos Marinos (LabQuiOM), Instituto Patagónico del Mar (IPAM), Facultad de Ciencias Naturales y Ciencias de la Salud, Universidad Nacional de la Patagonia San Juan Bosco, Boulevard Brown 2930, Puerto Madryn 9120, Argentina

**Keywords:** echinoderm welfare, 5Rs principle, respect, responsibility, invertebrates, responsible echinoderm use, echinoderm sentience

## Abstract

**Simple Summary:**

This study discusses the moral issues surrounding the study of echinoderms, a class of marine animals that is attracting interest from scientists. The significance of safely handling these animals is stressed and promotes an ethical standard that is customized to their needs. Our strategy was heavily influenced by the 3Rs principle, which advocates for substituting aware living vertebrates with non-sentient material in research. Echinoderms are excellent models for experimental inquiry because they are typically thought to be non-sentient. Although it is not possible to assess their mental states at this time, there is ample evidence of social behavior in many species, suggesting that ignoring interactions with them could be detrimental to their wellbeing. Recently, progress has been made toward developing an ethical framework for invertebrates, such as crustaceans, echinoderms, and cephalopods. To protect the welfare of echinoderms even in the absence of specific standards, we suggest an enlarged version of the 5Rs framework that includes responsibility and respect. In addition to advancing our knowledge of these interesting species, this research establishes a critical standard for the responsible and sympathetic care of all animals in scientific research.

**Abstract:**

This paper explores the ethical considerations surrounding research on echinoderms, a group of invertebrates that has recently garnered attention in the scientific community. The importance of responsible animal handling and the need for an ethical framework that encompasses echinoderms are emphasized. The 3Rs principle, advocating for the replacement of conscious living vertebrates with non-sentient material in research, is discussed as a guiding tool in current animal research practices. As invertebrates are generally classified as non-sentient animals, the replacement dimension tends to favor them as prevalent models in experimental research. While it currently lacks the means to assess the mental states of invertebrates, there is undeniable evidence of social behavior in many species, suggesting that a lack of interactions with these organisms could potentially adversely affect their wellbeing. In the last few years, considerable progress has been made in developing an ethical framework that takes invertebrates into account, particularly cephalopods, crustaceans, and echinoderms. In this context, we discuss the development of a broader conceptual framework of 5Rs that includes responsibility and respect, which may guide practices ensuring welfare in echinoderms, even in the absence of any particular normative.

## 1. Introduction

Animal use and interaction have played a crucial role in human endeavors throughout recorded human history. Food, transportation, research (primarily in the field of medical investigation), clothing, and companionship are among the main applications. Notably, this engagement includes invertebrates, indicating their natural involvement in various aspects of human existence. While some aspects of these relationships are unequivocally positive, such as providing sustenance, serving as research models, or providing companionship, others are negative, lacking in purpose, or even inflict harm. This dichotomy may stem from the classification of certain invertebrates as pests or vectors of human diseases. Such adverse interactions have elicited widespread aversion or apprehension toward numerous invertebrate species [1]. As a result, ethical concerns about these creatures must be addressed to ensure their viability as experimental modeling alternatives to vertebrates.

Invertebrates account for more than 90% of the total biodiversity on Earth [1]. This vast biological realm includes 36 phyla of invertebrates, 8 of which share the most common associations with humans: Porifera, Cnidaria, Platyhelminthes, Nematoda, Annelida, Arthropoda (the largest phylum in the animal kingdom), Mollusca (the second-largest phylum in the animal kingdom), and Echinodermata [1]. Although these phyla are all classified as invertebrates, their diversity is nothing short of astounding. Morphological, nervous system, and behavioral characteristics are unique to each phylum and can even vary within one [2,3,4].

Recently, there has been a shift in the ethical consideration of invertebrates, prompting a number of studies to lay the philosophical groundwork for incorporating invertebrates into ethical discourse [5,6,7,8]. The intricate behaviors of specific invertebrates, particularly octopuses (Cephalopods), have primarily prompted this transformation. Close interaction with octopuses in aquarium settings has aided in the identification of individuality and observable behaviors (personality traits) in these creatures [7,9]. Despite their distinct nervous system configuration from that of vertebrates, these studies have revealed octopuses’ remarkable intelligence and sentience [10]. Furthermore, in 2013, cephalopods were included in European Union legislation regarding the protection of animals used for scientific purposes, putting them on par with vertebrates (EU, 2010, Directive 2010/63/EU). This pivotal advancement not only adds to the ethical consideration for invertebrates in general, but it also represents a pivotal advancement in the field [3,8,11]. It is paramount to recognize that a lack of understanding of invertebrate behaviors does not preclude their capacity for sentience or their ability to respond to negative experiences in a non-anthropocentric manner that could cause suffering [12].

Concurrent with this growing ethical awareness, significant efforts have been made in experimental research to improve invertebrate welfare [5,7]. However, relevant information about the implications for invertebrate welfare is dispersed, limited, and sometimes contradictory. As a result, the current study conducts a comprehensive review of the current landscape of animal ethics and echinoderm welfare, with the goal of contributing to the development of a comprehensive framework for invertebrate welfare.

## 2. Ethical Concerns and the 5Rs Principle

In general, ethics is concerned with situations involving various types of conflicts. While there are many different types of conflicts in science, not all of them require ethical consideration. The most widely accepted criterion for determining the relevance of ethical reflection is whether or not it has an impact on humans [13]. This viewpoint clearly states that all human beings are morally concerned (an anthropocentric doctrine). However, the same cannot be said for non-anthropocentric living beings. As a result, determining which living beings should be regarded as ethical considerations becomes difficult.

One approach is to use humans as a reference point to determine whether these living beings meet criteria for sentience or consciousness, which include behavioral, evolutionary, and physiological considerations [14]. Scientific evidence supports arguments about which living beings should be considered ethically [6,12,15,16,17]. With our improved understanding of animal suffering, the scope of moral consideration has expanded to include all vertebrates, as well as octopuses as the sole group of invertebrates. Nonetheless, the ethical discourse is increasingly shifting toward considering animal species regardless of scientific knowledge about their capacity for suffering [18]. Significant efforts have recently been directed toward developing an ethical framework that takes invertebrates into account. Significant progress has been made in this regard for two distinct groups, namely crustaceans [19] and echinoderms [8,20].

The 3Rs principle, proposed by Russell and Burch (1959) [21], is the globally accepted tool that currently guides animal research practices. Replacement is one aspect of this principle that advocates for the use of scientific methods involving non-sentient material to replace those involving conscious living vertebrates [21]. Replacement should ideally encourage the use of lower levels of organization, such as cell culture, and even artificial models, such as computational simulations. However, because invertebrates are generally considered non-sentient animals, the replacement dimension tends to favor them as common models for experimental research. In this regard, the principle falls short in considering the inclusion of those invertebrates whose capacity for sentience is still debated. The 3Rs principle is built on a strong assumption: that non-human animals lower on the zoological scale lack sentience [22]. This assumption presents a significant limitation, especially considering the vast number and diversity of invertebrates used in scientific research. Despite the widespread acceptance of the 3Rs as a policy tool aimed at alleviating animal suffering and reducing their use in research, its effectiveness in achieving these goals has been notably criticized [23]. In fact, the number of animals used for research in the European Union (EU) is now similar to that in the 1980s [24]. The three R principle was originally intended to establish ethical guidelines for animal experimentation, albeit with species-specific limitations. However, it did not address the epistemological issues raised via animal manipulation adequately [25]. This principle was later expanded to the 5Rs principle (Figure 1) to include two additional concepts emphasizing personal commitment: respect (establishing a respectful relationship with any living being regardless of its complexity or the knowledge we have of that living being) and responsibility (personal commitment of researchers to apply ethics concepts conscientiously) [8].

## 3. Ethical Approach in Echinoderms

### 3.1. Phylum Echinodermata

Echinoderms are a phylum of marine invertebrates. They are often key, long-living species that shape and maintain the status of many marine ecosystems, inhabiting a wide range of ecological niches from the abyssal depths of the oceans to the intertidal zone. Echinoderms, distinguished by their pentamerous radial body arrangement, form a monophyletic group with hemichordates, also known as acorn worms. These organisms exhibit significant diversity and widespread distribution across a variety of marine habitats, playing critical ecological roles in each setting [26]. The fundamental characteristics of this phylum have exhibited remarkable consistency since the Ordovician epoch (approximately 495–440 million years ago) [27], with an approximate enumeration of 7000 species [28].

The extant echinoderm assemblage includes approximately 7000 species divided into five distinct taxonomic groups: Asteroidea, which includes starfishes; Echinoidea, which includes sea urchins, sand dollars, and sea biscuits; Crinoidea, which includes sea lilies and feather stars; Ophiuroidea, which includes basket stars and brittle stars; and Holothuroidea, which includes sea cucumbers. This collective lineage has an exceptionally notable fossil record dating back to the Cambrian period, providing a solid foundation for comparative molecular studies spanning a wide range of meticulously documented divergence intervals [27,28] (Figure 2).

Echinoderms have a number of distinguishing characteristics that set them apart from their zoological counterparts. Notably, among these characteristics is a hydraulic water vascular system that is intricately linked with a distinct arrangement of a calcium carbonate endoskeleton known as stereom. Most echinoderms’ developmental trajectories begin with larval stages, where intricate metamorphic processes culminate in the eventual manifestation of the adult form. These organisms exhibit a diverse range of life history traits, most notably sexual reproduction, though asexual reproduction is also observed. Echinoderm larvae are primitive, free-living, and planktonic in nature, with a wide range of morphology and functional characteristics that occasionally resemble those observed in hemichordate larvae. Following the metamorphic stage, the majority of mature echinoderms adopt a benthic lifestyle with radial symmetry and a typical pentameric structural composition. The organisms’ internal architecture is intricately organized, with a reliance on calcium carbonate ossicles reinforced by a complex network of collagenous ligaments. Notably, any skeletal structures found in larvae adopt intricate rod-like configurations with distinct origins in Ophiuroidea and Echinoidea, whereas such features are absent in the larvae of the remaining three echinoderm classes. Echinoderms have distinct pentaradial symmetry in their adult forms. Nonetheless, their developmental origins can be traced back to bilaterally symmetrical larvae, a shared feature of the deuterostome clade (shown in Figure 2A). These distinguishing characteristics place echinoderms within an intriguing evolutionary framework, identifying them as invertebrate deuterostomes inextricably linked to vertebrate organisms [29].

Echinoderm species exhibit a variety of developmental strategies, ranging from direct development from a fertilized egg to an adult to indirect development, in which adults emerge from the metamorphosis of a larva with no relation to the adult. The cell structure and count of indirect developing species’ long-living, feeding, bilaterally symmetrical larvae are very simple. There are numerous intermediate developmental stages, including facultative larval feeding and non-feeding larvae. Indirect development is primitive in echinoderms, and all five surviving classes, as well as the sister phylum Hemichordata, have dipleurula-type larvae [30].

Crinoids (Crinoidea) are sessile or free-living benthic animals with microphagous filter-feeding habits. Some comatulids feed during the day, while others feed by extending the tips of their arms or moving only at night to avoid predators. Some forms prefer and actively seek out areas with flowing water for feeding (rheophilic), while others do not (rheophobic). They can be found in a variety of habitats, including as attachments to substrates via their arms or cirri, within caves or small crevices, beneath rocks, or as attachments via cirri to other invertebrates such as corals (epizoic). Crinoids are important in developmental biology because they are the only echinoderms that have a primitive tripartite coelom. They are significant in paleontology because they evolved during the Cambrian period and were a dominant and diverse component of Paleozoic benthic fauna [31].

Sea cucumbers (Holothuroidea) are mostly found in benthic marine habitats ranging from the littoral zone to the abyss. They are found in all oceans, with particular abundance in coral reefs. Sea cucumbers are primarily detritivores, feeding on both suspended particles and organic matter bound in sediment. The distribution of sea cucumbers is influenced by the type of substrate, as well as other environmental variables such as dominant currents, temperature, water salinity, and depth. Sea cucumbers’ ecological importance is linked to their bioturbation activities, which involve the movement of organic material within sediments as well as the transfer of energy and materials at the water-sediment interface. The commercialization of sea cucumbers’ body walls, also known as beche de mer or trepang, is economically significant. Out of the over a thousand known species, the fishing industry primarily focuses on around thirty. This practice has a long history in Chinese and Japanese culinary and medical traditions. Larger species are especially valuable. Trepang curing is applied to the body wall, resulting in a product with high nutritional value due to its high protein and low fat content. Furthermore, sea cucumbers contain biologically active compounds that are used to treat a variety of medical conditions, including HIV, cancer, and osteoarthritis [32,33].

Echinoids (Echinoidea) are found in a wide variety of geographical and bathymetric environments, from intertidal zones to depths of 5000 m. Regular sea urchins live on a variety of substrates, with the majority of them living on rocky or mobile substrates. Sand dollars and heart urchins, for example, can only be found on soft bottoms and frequently bury themselves. Many morphological differences between regular and irregular echinoids are caused by differences in lifestyle and feeding habits. Some populations of regular sea urchins exposed to wave action have developed digging behaviors to bury themselves slightly, whereas species not exposed to such conditions typically exhibit cryptic behaviors [34]. Sea urchins, as well as sea cucumbers, are valuable species, and the commercialization of sea urchins’ gonads, also known as roe, is economically significant around the world [34,35].

Ophiuroids (Ophiuroidea) have a wide geographic distribution and a benthic lifestyle, having adapted to live in a variety of environments. They have been discovered in submarine hydrothermal vents and on bottoms with cold methane seeps, as well as in intertidal zones and abyssal regions from the tropics to the poles. The majority of species are usually found on soft substrates. They can be carnivorous, scavengers, sediment-consuming, or filter feeders. The majority use multiple feeding methods, though one usually takes precedence over the others. The majority are carnivorous, eating polychaetes, mollusks, and small crustaceans. Because of their diverse feeding strategies, they play an important role in trophic chains. They are not commercially important, unlike some other echinoderm groups. Nonetheless, due to their abundance, they can be used as environmental indicators [36].

Sea stars (Asteroidea) have a wide geographical range and a benthic habitat, inhabiting a variety of marine substrates where they can be abundant and visible. The vast majority are scavengers or opportunistic predators. Many species have generalist feeding habits and play an important role in the structure and functioning of marine communities as apex predators. They play an ecologically important role across latitudes by occupying different levels of the trophic chain, particularly as apex predators in rocky and coral reef ecosystems. This group’s ecological success can be attributed to a variety of morphological and life history characteristics, including indeterminate growth, extraoral and intraoral digestion (providing access to a diverse diet), rapid prey detection and response, and the ability to anchor themselves to substrates using ambulacral tube feet [36,37].

### 3.2. Historical Use of Echinoderms

Because sea urchins have been consumed by humans throughout history, the human understanding of this Phylum dates back to prehistoric times [38]. They are depicted in “frescos cretenses”, which date back 4000 years. For a long time, Eastern cultures have consumed and used sea cucumbers for medicinal purposes [39]. The earliest known of these animals dates from the period of Aristotle, who described the first known echinoderm in 350 B.C.E., over 2000 years ago. He described the feeding apparatus of sea urchins in his work “Historia Animalium”, which is now known as *Aristotle’s Lantern*. It is worth noting that Aristotle classified echinoderms as ostracoderms. The scholars of the time rekindled their interest in nature and began studying these creatures again. Klein coined the term “Echinodermata” in his 1734 work “Naturalis dispositio echinodermatum”. However, he only used it to refer to sea urchins, not all the classes that are now known. Linnaeus classified the genera Asterias, Echinus, and Holothuria as Mollusca in the 10th edition of *Systema Naturae* (1758). The term “Echinodermata” resurfaced in 1792, when it was recognized that these animals were a distinct group of invertebrates, though sea cucumbers were not included. Later, Lamarck (1809) [40] grouped Echinodermata with the true Coelenterata in the Radiata group of invertebrates. It took nearly four decades for De Tornos (1839) [41] and Salacroux (1840) [42] to coin the term “Echinodermata”. Due to its more advanced structural characteristics, it was argued in 1854 that Echinodermata did not belong with Coelelenterata. Echinodermata have since been recognized as a distinct clade of invertebrates.

Since the nineteenth century, the description of Echinodermata species has been a dominant focus in the literature. Initially, the focus was on species found along Europe’s coasts, as evidenced by Frey and Leuckart’s work in 1847 [43]. As a result of numerous oceanographic expeditions, this trend has expanded to include species from all over the world. While it is impractical to list every expedition and paper that resulted from it, some of the pioneering ones are worth mentioning. HMS Challenger and the Albatross laid the groundwork for hundreds of subsequent expeditions around the world. These expeditions have recently expanded to include deep-sea species as well as those found in intertidal and shallow waters [44,45,46,47,48]. Given that echinoderms have been collected worldwide for over 300 years to describe species, determine distributions, and populate museum collections, it is clear that hundreds of thousands of echinoderms were collected and preserved without much ethical thought.

### 3.3. Echinoderms as Models

Since the widespread use of the microscope in scientific research, echinoderms have served as experimental models. Due to the ease with which gametes could be obtained and the optical transparency of sea urchin embryos, they became valuable animal models in the mid-nineteenth century. Dufossé (1847) [49] and Derbès (1847) [50] provided early insights into fertilization and sea urchin embryo development via metamorphosis. Despite the fact that their work was overlooked in the scientific literature [51], sea urchin embryos remained useful as model organisms. Hertwig (1876) [52] made a pivotal discovery by demonstrating how sperm entered the female gamete, leading to embryo formation in the sea urchin *Toxopneustes lividus*. This discovery established sea urchin embryos as the gold standard for embryonic research, advancing our understanding of fertilization mechanisms, egg activation, cleavage, gastrulation, and early embryonic differentiation. They have also been useful in research on nervous system development, evolutionary development, and regeneration [51,53,54,55,56] (http://www.echinobase.org, accessed on 10 October 2023).

Sea stars were also recognized as important model organisms in the early twentieth century. Metchnikoff (1893) [57] discovered a cellular immune response in the bipinnaria larvae of sea stars, observing the rapid migration of mesenchymal cells to injury sites. He also discovered amoeboid cells’ phagocytic activities [57]. Despite these groundbreaking discoveries, echinoderm larvae were not widely used as model organisms in immunology until the twenty-first century (Figure 2A) [58,59,60].

One of the most remarkable characteristics of echinoderms is their ability to regenerate lost structures, such as the arms of sea stars. In recent years, there has been a surge in research into the regenerative capacity of the echinoderm nervous system (NS). Studies have shown that holothurians can regenerate their radial nerve cord (RNC) after it has been transected, with the regenerated cords displaying a similar structure and function to that of the originals and without scar tissue formation—a common issue in vertebrate CNS regenerative responses. Follow-up studies have revealed that radial glia play an important role in the regenerative process, both in forming the bridge that connects the severed ends of the RNC and in generating new neurons and glia in the regenerated structure [61,62]. Several laboratories are currently working to identify the genes required for NS regeneration [63,64,65,66,67].

The capacity for evisceration and subsequent regeneration in sea cucumbers, as well as brittle stars and sea stars due to their ability to regrow arms, has positioned them as invaluable models in regeneration research [66,68,69,70,71,72,73,74,75,76,77,78,79,80,81]. With recent technological advancements and the availability of new tools, echinoderms have become outstanding model organisms for both scientific research and educational purposes, and in most cases individuals are handled alive [82,83] (www.Echinobase.org, accessed on 10 October 2023).

### 3.4. Echinoderms Nervous System

In echinoderms, the nervous system is organized in accordance with the general pentameric pattern of the body plan. Each radius has its own radial nerve cord, which runs the length of the proximo-distal axis and terminates at the distal tip of the arm (in stellate echinoderms) or near the aboral pole (in globose forms). A circumoral nerve ring joins all five individual radial nerve cords at the body’s oral pole to form a single anatomical entity. The radial nerve cords and the nerve ring comprise the echinoderm’s central nervous system (CNS) (Figure 3). This CNS is an anatomically and histologically distinct agglomeration of neurons and glial cells associated with an extensive neuropil (densely interwoven neuronal processes) found nowhere else in the body and is in charge of the initiation and coordination of various body-wide responses [84].

Adult echinoderm neuroanatomy is distinguished by the presence of distinct superimposed domains or layers of nervous tissue located at different levels relative to the oral–aboral axis. These domains are known as the ectoneural and hyponeural systems [85,86,87]. The ectoneural system is located around the mouth, either within or directly beneath the oral epidermis. It is always present both in the nerve ring and radial nerve cords of all echinoderms, shows the most consistent organization across the phylum, and is the predominant part of the nervous system in all classes except crinoids. The hyponeural system may or may not be a part of the nerve ring or radial nerve cords. Its organization differs between classes and is generally related to the degree of development of large muscles. When hyponeural tissue is present, it acts as a second (usually thinner) layer of nervous tissue that directly overlies the aboral surface of the respective ectoneural cords [87,88].

This simplified classification of the echinoderm nervous system, however, does not fully account for recent reports from the last decade, revealing unexpected nervous system elements that do not neatly align with the three aforementioned divisions [64,84]. The nervous system (NS) of echinoderms is one of the most fascinating aspects. Given their close evolutionary relationship with chordates, as well as their radial symmetry and lack of prominent ganglia or centralized nervous structures associated with cephalization found in most other animals, the echinoderm nervous system has been regarded as pivotal to understanding the evolution of the chordate nervous system [29,84]. In fact, it has been suggested that the centralized nervous system is a plesiomorphic (ancestral) condition in echinoderms and may also be a plesiomorphic trait at the level of the Deuterostomia [84].

In the nervous system, including the ectoneural and hyponeural subsystems, tissue is organized as a neuroepithelium made up of two major cell types: radial glial cells and neurons. The cell types have a similar relative abundance in the radial nerve cord’s ectoneural and hyponeural bands, with radial glial cells accounting for 60–70% of the total cell population. On the other hand, neuronal cells are more abundant in the circumoral nerve ring, accounting for only 45% of the cell composition formed by glia [89]. Echinoderm glial cells share significant morphological similarities with the radial glia of chordates, including the orthogonal orientation of the cell’s main axis to the plane of the neuroepithelium, the presence of long thick bundles of intermediate filaments, and the presence of short protrusions branching off at a right angle from the main processes and penetrating into the surrounding neural parenchyma. The cell bodies of the majority of radial glial cells in echinoderms are located at the apical surface of the neuroepithelium. Some of them, however, are bipolar, with the apical and basal processes extending from opposite poles of cell bodies located at different depths within the neural parenchyma. Radial glial cells are the most common type of glial cell in echinoderms (though they are unlikely to be perfectly homogeneous), and they perform a variety of functions. In addition to the radial glia of the CNS, other glial cell types associated with the peripheral nervous system may exist [84]. Radial glia have more morphological similarities than do chordates because they are the main proliferative population in nervous tissue and thus capable of giving rise to new neurons, both in non-injured and regenerating CNSs. There are, however, differences between radial glia in echinoderms and chordates. Another significant difference is that radial glia are the only major glial cell type in the adult CNS of echinoderms, whereas in higher vertebrates, radial glia predominates in embryogenesis but then mostly disappear from the mature nervous system by giving rise to a plethora of more specialized cell types [90,91]. Radial glia are common in the adult CNS of lower vertebrates, but they frequently co-exist with other abundant specialized glia, such as those of oligodendrocytes [92,93].

The neural parenchyma of the CNS is made up of neurons with somata and neurites. The most common neuronal morphology ranges from unipolar and bipolar to multipolar. In echinoderms, neurons can be classified quite easily by size as normal neurons, which are dominant in most classes, and the giant neurons of ophiuroids. The first class has small somata (about 5 m in diameter) that produce very thin give processes (0.1–1 m in cross-section) with numerous local swellings (varicosities) along their length. Because they are typically found near or embedded within calcareous structures, they are less appealing to neurobiologists, particularly electrophysiologists who study their electrical properties. As a result, while other animal groups became popular neurobiology research subjects, echinoderms were largely overlooked for a long time. Even today, there are surprisingly few electrophysiological studies on echinoderms [94,95].

Communication between neurons in traditional chemical synapses was previously thought to be absent in echinoderms [85,96]. However, this long-held belief may be due to a lack of adequate tools for dealing with the difficulties imposed by the endoskeleton. Since the optimization of sample preparation protocols, there has been evidence demonstrating the presence of typical chemical synapses that occur on a regular basis in the CNS of Echinodermata [89,97]. These findings are consistent with those found in the sea urchin genome, where the genes required for synapse formation were discovered [83]. There are also different types of synapses: unsheathing synapses (with the pre-synaptic terminal wrapped around the post-synaptic process), passant synapses between parallel nerve fibers, and complex synapses with a pre-synaptic terminal forming two or more synapses in different post-synaptic processes, or, conversely, a single post-synaptic neuron receiving synaptic input from multiple pre-synaptic axons [97]. Echinoderm CNSs have regional differences in cellular composition as well as a complex internal spatial segregation of different cell types. The radial nerve cord is made up of repetitive units [85,98,99].

The CNS of echinoderms generates complex, coordinated, and directional behavioral responses to various sensory stimuli. Although the molecular and cellular mechanisms underlying these behaviors remain unknown, it is known that an echinoderm’s CNS contains a large number of neurotransmitters from all major groups, including acetylcholine, aminoacids, monoamines, neuropeptides, and gases. Acetylcholine appears to mediate muscle contraction due to its function as a major excitatory neurotransmitter [88,100,101]. Post-synaptic nicotinic and muscarinic receptors have also been identified, and acetylcholinesterase (the enzyme required to hydrolyze acetylcholine at synapses) activity has been detected in both ectoneural and hyponeural systems [85,102]. GABAergic neurons proliferate throughout the CNS, including the radial nerve cord, nerve ring, and podial nerves, as well as the nerves and visceral plexi [84,103]. GABA is involved in both echinoderm muscle contraction and relaxation, depending on the post-synaptic receptor (GABA A or GABA B) present in neuromuscular junctions [100]. L-glutamate is and excitatory neurotransmitter in the ectoneural subsystem of echinoderms [104,105]. L-glutamate is also a neurotransmitter capable of eliciting the arm autotomy response, whereas acetylcholine acts as an antagonist of L-glutamate [104]. The research on serotonin as a neurotransmitter in the adult CNS is limited. There have been reports of its presence in muscles and basiepithelial plexi [106,107,108]. Furthermore, neuropharmacological studies have revealed that serotonin regulates muscular contraction by inhibiting the excitatory effect of acetylcholine [107] and may also be involved in the regulation of post-traumatic regeneration [108]. Catecholamines such as dopamine and noradrenaline have been found in the ectoneural only and appear to be involved in the movement of tube feet [109,110] and to be fundamental for the righting response of the sea urchin [111]. Histamine data of the sea urchin are extremely limited; they have only been studied in one species of cucumber, and the latter appears to be involved in sensory systems, as it was found in tentacles and body wall papillae, and to project its axons directly to the nerve ring and the radial nerve cord [112].

The enzyme nitric oxide synthase, which produces NO, was discovered in both the ectoneural and hyponeural parts of the radial nerve cord of adult sea stars, as well as in some radial glial cells [113,114]. Apparently, NO is involved in the relaxation of viscera and tube feet [115,116].

In addition to these phylogenically widespread neurotransmitters, echinocherms contain specific neuropeptides from the SALMFamide family [88,98,116,117,118,119]. These neuropeptides relax the visceral musculature as well as the muscles of the body wall [116,120].

All of the information above suggests that the echinoderm CNS is more complex than previously thought and that, despite the lack of a centralized brain, it is possible for it to elicit complex individual and social behavior. Recent advances in knowledge of this group have provided new insights suggesting that echinoderms are sentient animals capable of suffering pain.

### 3.5. Pain and Echinoderms

Even though the debate over pain perception in invertebrates is still ongoing, it is critical to recognize that the absence of evidence of painful sensations should not be interpreted as conclusive proof of pain absence in this group. As scientists, we are responsible for treating the organisms we study with dignity and ensuring their wellbeing. To accomplish this, we must first understand the concepts of analgesia, sedation, and anesthesia. It is also critical to understand how to correctly administer anesthesia, which response variables to consider, and which substances to use.

Various agents have been used to anesthetize echinoderms [121]. Iso-osmotic solutions such as MgCl_2_, MgSO_4_, or Ca^2+^-free seawater are common echinoderm anesthetics. The mechanism of action of these agents is to destabilize membrane potential, preventing pain signals from propagating. Additionally, local anesthetics that block neural stimulation of the muscle have been employed when necessary. The local anesthetics MS222 [119,122,123,124,125] and propylene phenoxetol [122,126,127,128,129,130,131] have both demonstrated effectiveness in studies involving echinoderm connective tissue.

MS-222 (IUPAC name 3-amino benzoic acid, ethyl ester, and methanesulfonate salt, also known as ethyl m-aminobenzoate or tricaine methanesulfonate) is a local anesthetic of the ester type. Its structure is similar to that of other local anesthetics, such as benzocaine, implying that it likely functions similarly by impeding axonal conduction via interference with membrane depolarization [132]. Originally developed as a fish anesthetic [133], it has since been widely used on a variety of invertebrates (National Research Council, 1981).

In ophiuroids, “propylene phenoxetol” is thought to act as a local anesthetic that inhibits axonal conduction [131]. Because this volatile liquid is not water-soluble, maintaining known concentrations of the compound in a medium is difficult. Furthermore, there have been some doubts about the precise identity of this compound. While certain compounds possessing anesthetic effects in echinoderms have been studied, the practice of using them during individual manipulations is not yet standardized because there is neither awareness nor a normative mandate for it.

## 4. Echinoderm Welfare

Animal welfare is now recognized as a scientific discipline that includes ethology, physiology, pathology, biochemistry, genetics, immunology, nutrition, cognitive–neural studies, veterinary care, and ethics [134,135,136,137,138,139,140]. While the assessment of animal welfare has traditionally focused on vertebrates [141], the vast diversity of invertebrates presents a significant challenge.

Firstly, the fundamental indicators (such as cortisol levels, longevity, feeding rate, and behavior) utilized in welfare assessments follow a reductionist approach. Secondly, this assessment only encompasses two of the “Three Conceptions” (basic health and function, and natural living) and four of the “Five Domains” (nutrition, environment, health, and behavior) that should ideally be evaluated. By definition, this basic assessment remains incomplete. Moreover, in invertebrates, except for cephalopods, evaluating the mental state domain and the conception of affective state is currently extremely challenging. Although we lack the tools to assess invertebrate mental states, there is undeniable evidence of social behavior in many species, implying that a lack of interactions with these organisms may have a negative impact on their mental wellbeing [142]. Even if a comprehensive welfare assessment for invertebrates is not currently feasible, we must ensure that the best welfare conditions are met.

A comprehensive assessment of animal welfare, according to Botreau et al. (2007) [143], requires a well-defined set of criteria. The following guidelines should be followed when developing these criteria:Each and every significant aspect must be addressed in order for the assessment to be comprehensive;The criteria must not be redundant or irrelevant;Each criterion must be independent of the others;All stakeholders must agree on the criteria, and they must have a practical basis;The criteria, as well as their application, should be transparent and simple to understand;The number of criteria should be limited to 12 at most.

Given the diversity of invertebrates, it is critical to recognize the importance of developing a specific set of criteria for assessing the welfare of each phylum or even each order within this group. There are several indicators that provide information about an animal’s wellbeing in response to its experiences. These indicators are based on a thorough understanding of how individuals respond physiologically and behaviorally to various conditions. The interpretation of these indicators is dependent on one’s attitude toward animal welfare. Exploration, hunting or foraging, socialization, parental care, play, and sexual activity are all expected to develop as inherent behavioral elements [144,145]. As a result, when an animal is kept in captivity, it is critical to understand the specific fundamental characteristics of each species that exist in their natural environment in order to recreate them. This meets the need to “explore, solve problems, and overcome challenges” [146] (p. 623).

In order to achieve the application of echinoderm wellbeing, the following non-invasive echinoderm assessments have recently been developed as indicators for use in the aquaculture industry and research, providing valuable insights into individual wellbeing.

Tube feet adhesion: Echinoderms move by using their tube feet, which are frequently adhesive to the bottom or walls of an aquarium in captivity. A detached or loosely attached echinoderm indicates that the individual is not in good health.Echinoderms have distinct righting behavior because they are most vulnerable with their oral face above [147,148]. The speed at which this behavior is carried out can be used to gauge their physiological activity, health, and overall condition. This indicator must first be established for the species, and size and sex must be evaluated before establishing a normal time of righting for each species. After you have set a time, you can use it as a stress indicator.Echinoderms use their spines/pedicelaria/arms for a variety of functions (feeding, moving, defense, and eating), so their response to stimuli is a good indicator of their health. An echinoderm that is nonresponsive or responds slowly indicates that the individual is not in good health.Feeding behavior: like any other animal’s, the feeding behavior of echinoderms is an indicator of their health. Echinoderms feeding and defecating indicate healthy individuals.Epidermis appearance: A healthy echinoderm has a shiny, non-interrupted epidermis. In contrast, the presence of reddish or blackish coloration, as well as inflammation and mucus, indicates the presence of an infection.

## 5. Conclusions

The majority of ethical and welfare approaches in animal research have primarily focused on vertebrates. Addressing invertebrate welfare presents unique challenges, and researchers will require time to fully integrate these concepts. However, significant progress in this area has been made. With rising public awareness and concern, there is a chance that ethical concepts will be adopted more quickly. In the near future, this could lead to the establishment of guidelines, norms, and laws in this domain.

In recent years, it has become clear that advanced invertebrates have qualities such as self-awareness and sentience, as well as the ability to experience pain, though precisely defining and understanding it in their context is difficult. Animal welfare legislation in various countries supports this viewpoint. While many invertebrates have learning and memory abilities, there are significant structural and physiological differences between animal groups. Some researchers believe that these differences indicate that, despite their self-protective behaviors, advanced invertebrates are incapable of feeling pain. Regardless of the validity of this argument, it is critical for humanity, particularly scientists, to take invertebrate welfare seriously and to treat them with care, both in captivity and in their natural habitats. The scientific community has responded by advocating for invertebrate welfare to be considered in breeding or holding facilities, laboratories, and under natural conditions whenever possible. In addition, if advanced invertebrate experimentation is deemed necessary, appropriate anesthesia methods should be considered.

This research provides evidence of sentience within the echinoderm group, highlighting important advances in our understanding of their biology and physiology. However, there are still notable gaps in information that require further investigation. Regardless of the current state of knowledge regarding sentience in echinoderms, there is a growing moral consideration to implement practices ensuring animal welfare when utilizing individuals in research. The 5Rs principle provides a useful conceptual framework in this situation by highlighting the importance of Replacement, reduction, and refinement as well as respect and responsibility in researchers’ practices while working with echinoderms.

## Figures and Tables

**Figure 1 animals-13-03377-f001:**
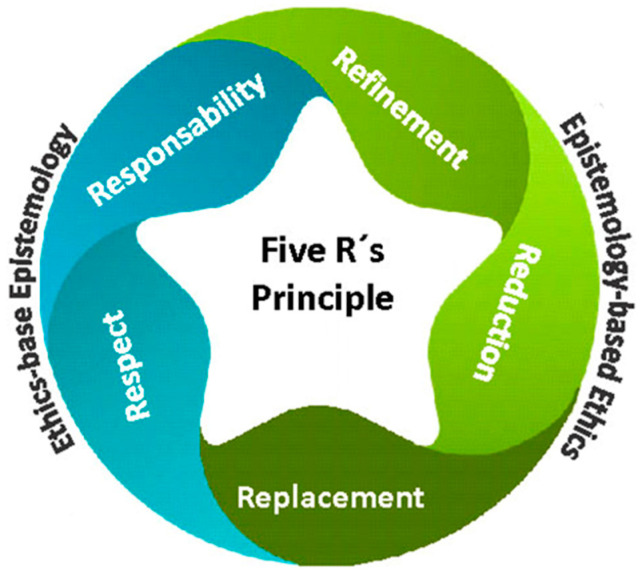
Five Rs principle which balances the level of importance of the ethical and epistemological approaches [8].

**Figure 2 animals-13-03377-f002:**
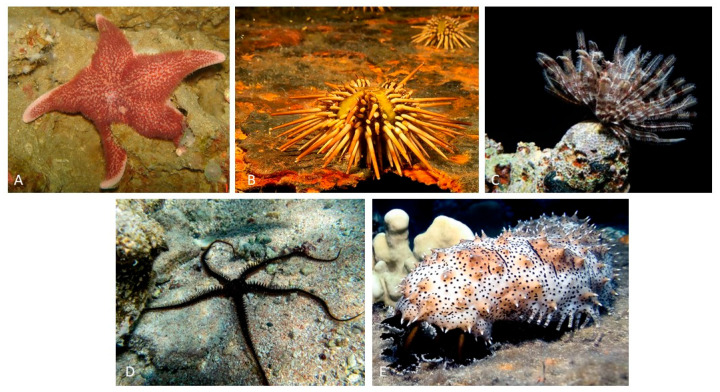
Example of classes of echinoderms: (**A**) *Cycethra verrucosa*, Asteroidea from South Atlantic; (**B**) *Arbacia dufresnii*, Echinoidea from South Atlantic; (**C**) *Ctenantedon kinziei*, Crinoidea from the Caribbean; (**D**) *Ophiocoma echinata*, Ophiuroidea from the Caribbean; (**E**) *Pearsonothuria graeffei*, Holothuroidea from the Indo-Pacific.

**Figure 3 animals-13-03377-f003:**
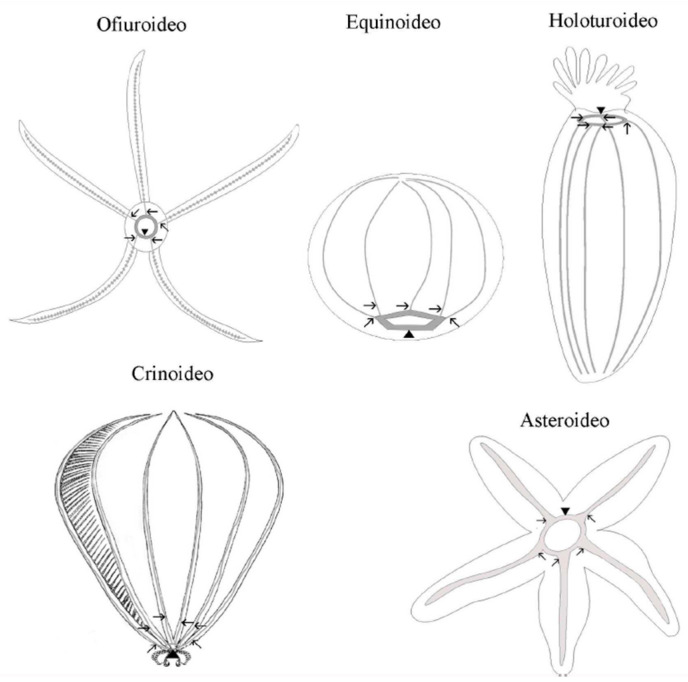
Schematic representation of the central nervous system of the phylum Echinodermata. The arrows indicate the radial nerves cords and the point of the arrows indicates the circumoral ring.

## Data Availability

Not applicable.

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
