# Peer review of "Ethical Considerations for Echinoderms: New Initiatives in Welfare"

_animals, 2023, doi:10.3390/ani13213377_

Round 1

Reviewer 1 Report

Comments and Suggestions for Authors

In the review “Ethical Considerations for Echinoderms: New Initiatives in Welfare” by Crespi-Abril A. and Rubilar T.

The authors in this review address the topic of animal welfare in invertebrates, in particular in echinoderms.

This is a topic of great interest to the scientific community and is addressed comprehensively here.

The conclusions require revision, this paragraph should refer to what is stated in "2. Ethical concerns and the 5R principle". The authors should clarify whether Respect and Responsibility can also be applied to the use of echinoderms.

Author Response

We revised the conclusion and we included how the 5Rs principle should be considered in echinoderms, as suggested by the reviewer. 

Reviewer 2 Report

Comments and Suggestions for Authors

I recommend minor revision due to some editorial points. All my remarks are present in the text of MS. To see all of them, please open the file in Acrobat Reader.

Author Response

We modified all the points indicated in the pdf file. 

Reviewer 3 Report

Comments and Suggestions for Authors

This paper discusses the moral issues surrounding in studying the study of echinoderms (Crinoidea, Holothuroidea, Echinoidea, Ophiuroidea and Asteroidea), introducing their living environment, developmental processes, morphological characteristics, and pain perception.

Since social and other behaviors have now been observed in some invertebrates, it is very important to consider whether ethical considerations should be made for invertebrates.

I would like to make a few comments.

1. The figure needs a detailed legend. Is there a species name for Fig.2? In which ocean are these species major?

2. In fig3, please clarify what the arrows and arrowheads are pointing to.

3. There is a description of the nervous system and pain sensation. It mainly mentions anesthesia, are there any previous studies that discuss how to verify the presence or absence of pain sensation?

Author Response

In this revised version of the manuscript, we have incorporated the adjustments to the captions for Figures 2 and 3 as recommended by the reviewer. Regarding the query about studies addressing the assessment of pain perception in echinoderms, the current emphasis lies in identifying the presence of nociceptors. In our paper, we consolidate the findings in this area to offer a comprehensive overview of the existing knowledge. It's important to note that the discussion on echinoderms' sentience is still in its early stages, and we have included the most recent papers on this subject for a more thorough examination.